# Enteric Neural Network Assembly Was Promoted by Basic Fibroblast Growth Factor and Vitamin A but Inhibited by Epidermal Growth Factor

**DOI:** 10.3390/cells11182841

**Published:** 2022-09-12

**Authors:** Jeng-Chang Chen, Wendy Yang, Li-Yun Tseng, Hsueh-Ling Chang

**Affiliations:** 1Department of Surgery, Chang Gung Children’s Hospital, College of Medicine, Chang Gung University, Taoyuan 333, Taiwan; 2Graduate Institute of Clinical Medicine, College of Medicine, National Taiwan University, Taipei 100, Taiwan; 3Pediatric Research Center, Chang Gung Children’s Hospital, College of Medicine, Chang Gung University, Taoyuan 333, Taiwan

**Keywords:** basic fibroblast growth factor, epidermal growth factor, vitamin A, enteric neural stem cell, gangliogenesis, neural network, neurosphere

## Abstract

Extending well beyond the original use of propagating neural precursors from the central nervous system and dorsal root ganglia, neurosphere medium (NSM) and self-renewal medium (SRM) are two distinct formulas with widespread popularity in enteric neural stem cell (ENSC) applications. However, it remains unknown what growth factors or nutrients are crucial to ENSC development, let alone whether the discrepancy in their components may affect the outcomes of ENSC culture. Dispersed enterocytes from murine fetal gut were nurtured in NSM, SRM or their modifications by selective component elimination or addition to assess their effects on ENSC development. NSM generated neuriteless neurospheres, whereas SRM, even deprived of chicken embryo extract, might wire ganglia together to assemble neural networks. The distinct outcomes came from epidermal growth factor, which inhibited enteric neuronal wiring in NSM. In contrast, basic fibroblast growth factor promoted enteric neurogenesis, gangliogenesis, and neuronal wiring. Moreover, vitamin A derivatives might facilitate neuronal maturation evidenced by p75 downregulation during ENSC differentiation toward enteric neurons to promote gangliogenesis and network assembly. Our results might help to better manipulate ENSC propagation and differentiation in vitro, and open a new avenue for the study of enteric neuronal neuritogenesis and synaptogenesis.

## 1. Introduction

Neurosphere medium (NSM) [1] and self-renewal medium (SRM) [2,3] employed to cultivate enteric neural stem cells (ENSCs) were developed from scratch and fundamentally differed in their components. NSM was prepared on the basis of Dulbecco’s Modified Eagle Medium/Nutrient Mixture F-12 (DMEM/F12) and supplemented with epidermal growth factor (EGF), basic fibroblast growth factor (bFGF), N2 and B27 supplements. Its prototype could be traced back to the protocol formulated for stem cells of the central nervous system (CNS) by Weiss’s group in 1986, consisting of DMEM/F12 and the supplements of N2-equivalent hormone and salt mixture in place of serum [4]. Later, this serum-free formula was supplemented with EGF [5,6] or bFGF [7]. Then, B27 was added to NSM for rapid and long-term growth of CNS precursors [8]. Since 2007, NSM or its relevant modifications had been widely used by Metzger et al. [9,10,11] and many other groups [1,12,13,14] to generate enteric neurospheres. With regard to SRM, its prototype was first put forth by Anderson’s team in 1999 for self-renewal and differentiation of post-migratory neural crest stem cells (NCSCs) from dorsal root ganglia (DRG) [15] as a low-glucose DMEM-based formula supplemented with chicken embryo extract (CEE), bFGF, N2, B27, and retinoic acid (RA). Bixby et al. appended insulin-like growth factor 1 (IGF-1) to Anderson’s NCSC formula to promote the NCSC survivals without affecting their differentiation [16]. In 2003, enteric neurospheres were first generated from dispersed enterocytes (referring to all the cells dissociated from intestine tissues) in this NCSC medium [17]. Later, Bixby’s formula was supplemented with neurobasal medium (NBM) and the term SRM was coined to describe its robust capacity to support self-renewal of murine NCSCs [3] or ENSCs [2].

Although NSM and SRM have extended well beyond their original use of propagating CNS and DRG precursors, and enjoyed widespread popularity in ENSC applications [12,13,18,19], nothing has been known about whether ENSCs shared neurotrophic factors and essential nutrients in common with CNS or DRG precursors given the difference in their embryological origin or developmental pathways of NCSCs [2,15,20]. In comparison with NSM, SRM necessitates additional supplements of CEE, RA, NBM, and IGF-1, but lacks EGF. However, the composition discrepancy between NSM and SRM did not reportedly bring any influences on the outcomes of ENSC culture, raising the question of whether these additional supplements or EGF were crucial to ENSC propagation. A better understanding of nutritional or neurotrophic requirements for ENSC development may help in manipulating ENSC proliferation and differentiation in vitro. It prompted us to assess the effects of NSM and SRM components on ENSC development. We disclosed that bFGF and RA helped in assembling enteric ganglia and neural networks in culture systems, whereas EGF precluded enteric ganglia from wiring together.

## 2. Materials and Methods

### 2.1. Mice

Inbred FVB/N mice were purchased from the National Laboratory Animal Center (Taipei, Taiwan) at the age of 6–8 weeks. Animals were housed in Animal Care Facility at Chang Gung Memorial Hospital (CGMH) under the standard guidelines from “Guide for the Care and Use of Laboratory Animals” and with the approval (IACUC2019091205 and 2021092201) of CGMH Committee on Animal Research. Females were caged with males in the afternoon and checked for vaginal plugs the following morning. The day of the plug detection was designated as day 0 of the pregnancy.

### 2.2. Harvest of Intestine in Fetal Mice

Under anesthesia with ketamine (100 mg/kg; Pfizer, NY City, USA) and xylazine (10 mg/kg; Bayer, Leverkusen, Germany), pregnant mice were subjected to midline laparotomy to expose uteri on their gestational day 14–15. All the fetuses were delivered through hysterotomy and immediately washed with saline. Following decapitation and laparotomy, fetal intestine was harvested and bathed in saline.

### 2.3. Culture of Enteric Neurospheres

Fetal gut was incubated in 1 mg/mL Collagenase/Dispase (Roche, Mannheim, Germany) in phosphate-buffered saline (PBS) for 15–30 min at 37 °C. Digested tissue was triturated and washed. Then, dissociated enterocytes of 2 × 10^5^/well were non-adherently grown in 6-well culture plates (TPP 92006; Trasadingen, Switzerland) flooded with 1.2 mL NSM, SRM or their relevant modifications. Serum-free NSM (100 mL) was prepared on the basis of 96 mL DMEM/F12 supplemented with EGF (20 ng/mL; PeproTech, London, UK), bFGF (20 ng/mL; PeproTech), 1 mL N2 (Gibco), 2 mL B27 (Gibco), and 1 mL penicillin-streptomycin (Gibco). Standard SRM of 100 mL was prepared by 50 mL low-glucose DMEM, 30 mL NBM (Gibco, Thermo Fisher Scientific, Waltham, MA, USA), 15 mL CEE, bFGF (20 ng/mL), IGF-1 (20 ng/mL; PeproTech), 1 mL N2, 2 mL B27, 100 µL RA (117 µM, Sigma-Aldrich, St. Louis, MO, USA), 1 mL penicillin-streptomycin, and 100 μL 2-mercaptoethanol (50 mM; Gibco). For component modifications, one component was eliminated from NSM or SRM. The medium volume deficit was replenished by its basal media of DMEM/F12 or low-glucose DMEM. Culture medium was replaced every 3–4 days.

### 2.4. Enumeration of Enteric Neurosphere Formation

Dissociated enterocytes were cultivated in SmartDish™ 6-well plates (STEMCELL Technologies, Vancouver, BC, Canada) as described above for 1 week. Then, neurospheres were enumerated under the 10× objective. STEMgrid™-6 counting grid (STEMCELL Technologies) was attached to the bottom of SmartDish™ 6-well plates. Five areas, of which each contained 4 grids of 2 mm × 2 mm, were marked for sphere enumeration. In each well, neurospheres generated on 20 selected grids were counted and classified by their maximal diameter of <50 μm, 50–100 μm, and >100 μm. Enteric neurospherogenesis was defined by the total neurosphere number per unit area (mm^2^).

### 2.5. Flow Cytometric Analyses for Neurons and ENSCs [21]

Colonies in culture wells were dissociated by pre-warmed (37 °C) 0.025% trypsin for 4 min, and then quenched by adding twice the volume of flow buffer (2% fetal bovine serum in PBS). Cell suspension was gently triturated by a microliter pipet to prepare a single cell suspension. Following PBS wash and centrifugation, cell pellets were resuspended in an appropriate volume for intracellular staining. Cells were fixed and permeabilized by BD Cytofix/Cytoperm^TM^ Kit (BD Biosciences, Franklin Lakes, NJ, USA). Then, samples were subjected to staining with anti-Tubulin β3 (TUBB3, TUJ1, BioLegend, San Diego, CA, USA) or anti-p75 neurotrophin receptor (p75, expressed in ENSCs [16,18,19,22], ab8875, Abcam, Cambridge, UK) primary antibodies, followed by appropriated secondary antibodies conjugated with fluorescence. Following washing, cells were acquired by BD FACSCantoTM II and analyzed with BD FACSDiva software.

### 2.6. Immunofluorescence Staining for Enteric Neurons and Gliocytes

Cells grown on coverslips were first subjected to fixation and permeabilization with BD Cytofix/Cytoperm^TM^ Kit. Then, samples were incubated with primary antibodies against TUBB3 and glial fibrillary acidic protein (GFAP, Poly28400, BioLegend), followed by fluorescence-conjugated secondary antibodies of rat anti-mouse IgG_2a_ (BioLegend) and donkey anti-rabbit IgG (BioLegend). Counterstain for the nuclei was performed using diamidino-2-phenylindole (DAPI, Sigma-Aldrich). After washing, specimens were mounted with Dako fluorescence mounting medium (Agilent, Santa Clara, USA) and examined under a Leica confocal microscope.

### 2.7. Neural Network Assembly from Enteric Neurospheres

Enteric neurospheres generated in NSM were seeded on fibronectin-coated (human fibronectin of 10 μg/mL for 60 min; Sigma-Aldrich) coverslips placed in culture wells flooded with NSM, EGF-free NSM (NSM deprived of EGF), CEE-free SRM, and EGF-supplemented CEE-free SRM. On day 7, neural network assembly was photographed live and after Diff-Quik staining. The neuroconnectivity in the network assembled from a single neurosphere was graded by the number of network nodes (intersections of neurite bundles) [13] counted after Diff-Quik staining.

### 2.8. Statistical Analyses

All numerical data were shown in box plots. The equality of means was examined by Student’s *t*-test between two independent groups or by one-way analysis of variance (ANOVA) among three groups with post-hoc Fisher’s least significant difference (LSD) multiple comparisons. Differences were regarded as significant in all tests at *p* < 0.05.

## 3. Results

### 3.1. Formation of Enteric Neurospheres in NSM and SRM

Dispersed enterocytes were first nurtured non-adherently in NSM and SRM, respectively for a baseline assessment of their neurospherogenic outcome. It was found that both protocols could lead to the generation of enteric neurospheres. Notably, NSM generated neurospheres with the scarcity of neurite outgrowth, whereas SRM contributed to the assembly of enteric neural networks (Figure 1A) with ganglion-like (containing neurons and gliocytes) neurospheres wired together by neurite bundles (Figure 1B).

### 3.2. Assessing Component Effects on Enteric Neurospherogenesis in NSM

DMEN/F12-based NSM was supplemented with bFGF, EGF, N2, and B27 as a type of serum-free and chemically-defined nutrient medium. It permitted us to investigate the nutritional or neurotrophic requirements for neurospherogenesis (neuriteless gangliogenesis) by selective component elimination from NSM. Enteric neurospherogenesis was blocked following the deprivation of bFGF or B27, but not affected by the exclusion of EGF or N2 supplement from NSM (Figure 2A). It indicated the vital role of bFGF and B27 in the assembly of enteric neurospheres.

### 3.3. Assessing CEE Effects on Neural Network Assembly in SRM

Low-glucose DMEM-based SRM was supplemented with the serum analogue of CEE, the nutritional supplements of NBM, RA, B27, and N2, and the growth factors of bFGF and IGF-1. Given that neural networks developed in SRM rather than NSM, it called into question whether the biological fluid of CEE was crucial to neuronal wiring. Therefore, we examined whether the omission of CEE from SRM affected neural network assembly in vitro. CEE-free SRM (SR1) maintained the comparable fractions of p75^+^ ENSCs and TUBB3^+^ neurons to original CEE-supplemented SRM (SR0, Figure 2B) in resulting cell compositions after the 7-day culture, and retained the capacity to assemble enteric ganglia (Figure 2C) and neural networks (SR1, Figure 2D). Therefore, CEE was a superfluous supplement to neuronal differentiation and wiring in SRM.

### 3.4. Assessing Component Effects on Neural Network Assembly in CEE-Free SRM

CEE-free SRM (SR1) consisted of chemically-defined constituents, which permitted us to experimentally explore the relevance of nutrients or growth factors to neural network assembly. The selective exclusion of IGF-1 (SR3), NBM (SR7), RA (SR9) or N2 (SR10) supplement from CEE-free SRM (SR1) did not disturb neurogenesis (Figure 2E), nor did it affect gangliogenesis (Figure 2C) and neuronal wiring (Figure 2D). However, the selective elimination of bFGF (SR2) or B27 (SR8) from CEE-free SRM (SR1) significantly lessened neurogenesis (Figure 2E) and gangliogenesis (Figure 2C) to preclude neural network assembly (Figure 2D), highlighting the indispensability of bFGF and B27 to ENS network assembly.

To reappraise the neurotrophic effects of IGF-1 and bFGF on ENSCs, we prepared IGF-1/bFGF/CEE-free SRM (SR6) that was devoid of IGF-1 and bFGF. Sparse enteric gangliogenesis and neuronal wiring in SR6 could be enhanced by bFGF rather than IGF-1 supplementation (SR3 and SR2, respectively, Figure 2C,D), while the difference in neurogenesis between SR3 and SR6 did not reach statistical significance (Figure 2E). Moreover, bFGF addition to EGF-supplemented but IGF-1/bFGF/CEE-free SRM (SR5) promoted neurogenesis (SR4, Figure 2E) and gangliogenesis (SR4, Figure 2C) despite sparse neurite outgrowth (SR4, Figure 2D). Together, the results strengthened the role of bFGF rather than IGF-1 in facilitating ENSC neurogenesis, gangliogenesis, and network assembly.

### 3.5. The Influence of EGF on Neural Network Assembly

EGF was not an essential ingredient for SRM, but rather a requisite for NSM. Enteric neurospherogenesis made no distinction as to whether NSM was supplemented with EGF or not (Figure 2A). However, EGF supplementation to IGF-1/CEE-free (SR3) and bFGF/IGF-1/CEE-free SRM (SR6) not only lessened neurogenesis (SR4 and SR5, respectively, Figure 2E), but also inhibited gangliogenesis (SR4 and SR5, respectively, Figure 2C) and neuronal wiring (SR4 and SR5, respectively, Figure 2D). Therefore, EGF precluded ENSCs from neurogenesis, gangliogenesis, and network assembly.

The influences of EGF on neural network assembly from enteric neurospheres were further investigated in NSM, EGF-free NSM, CEE-free SRM, and EGF-supplemented CEE-free SRM. EGF omission from NSM caused neural network assembly (Figure 3A), whereas EGF supplementation to CEE-free SRM inhibited neuronal wiring (Figure 3B). The neuroconnectivity in the assembled networks was quantified by the number of neurite intersections in the networks. The presence of EGF in media significantly lessened the neurite intersections of the network assembled from a single neurosphere (Figure 3C), reflecting sparse connectivity in neural circuits. EGF supplemented to CEE-free SRM also blocked neurite extension and neuroconnectivity between two individual neurospheres (Figure 3D). Therefore, EGF precluded enteric neurospheres from neurite outgrowth and wiring. These results further strengthened the inhibitory role of EGF in ENS network assembly.

### 3.6. Reappraisal of RA Effects on Neural Network Assembly

Selective RA elimination from CEE-free SRM did not affect ENS network assembly. Given that B27 supplement contained vitamin A analogue of retinyl acetate [23], it might be premature to conclude the irrelevance of vitamin A to enteric neuronal wiring. As we know, vitamin A is a group of unsaturated nutritional organic compounds including preformed vitamin A (retinol and its esterified derivatives), retinal, and provitamin A carotenoids [24]. Both preformed vitamin A and provitamin A can be intracellularly metabolized to retinal and then RA (the active forms of vitamin A) to support the biological functions relevant to visual cycle, receptor binding, and the regulation of target gene expression [24]. Therefore, RA deprivation from CEE-free SRM (SR1) was unable to completely block biological pathways of vitamin A in RA/CEE-free SRM (SR9) given that the upstream retinyl acetate in B27 supplement was functionally equivalent to RA.

Based upon SR9 formula, we substituted B27 supplement minus vitamin A (B27-VA, without retinyl acetate) for the original B27 supplement. The complete absence of vitamin A derivatives in the B27-VA SR9 variant caused a significant loss of gangliogenic capacity and arrested neuronal wiring, whereas RA addback to B27-VA SR9 restored the assembly of ganglia and neural networks (Figure 4A,B). Notably, complete deprivation of vitamin A derivatives had no substantial influence on the fractions of p75^+^ ENSCs and TUBB3^+^ neurons (Figure 4C). Further cytometric analyses in a two-dimensional plot revealed that complete vitamin A deprivation retarded neuronal maturation with TUBB3^+^ neuron stagnation in a state of p75^high^ expression during p75^+^ ENSC differentiation along a neuronal lineage [25] (Figure 4D). RA addback resumed the differentiation of TUBB3^+^p75^high^ neurons toward TUBB3^+^p75^low^ or TUBB3^+^p75^−^ neurons (Figure 4D,E). Therefore, vitamin A derivatives might facilitate neuronal maturation in the process of neuronal differentiation, and promote the assembly of enteric ganglia and networks.

### 3.7. The Influence of Glucose Concentration on Neural Network Assembly

SRM was made of DMEM with 1.0 mg glucose/mL from the outset [15]. To examine the influences of glucose concentration on neuronal wiring, we substituted high-glucose (4.5 mg/mL) for low-glucose (1.0 mg/mL) DMEM in SRM and further subjected the high-glucose SRM to selective component eliminations as performed in original low-glucose SRM (Figure 2C). The assembly of enteric ganglia (Figure 5A) and networks (figures not shown) in high-glucose SRM benefitted significantly from bFGF and B27 rather than CEE, IGF-1, NBM, RA, and N2, essentially in line with what was observed in low-glucose SRM. However, high-glucose CEE-free SRM provided a higher yield of enteric ganglia (mainly sphere size >100 µm, Figure 5B,C), p75^+^ ENSCs, and TUBB3^+^ neurons (Figure 5D) than its low-glucose counterpart.

## 4. Discussion

Enteric nervous system (ENS) develops as a consequence of the migration, proliferation, and differentiation of the neural crest-derived ENSCs [26]. Nevertheless, little is known about what serum or humoral factors regulate or control ENSC proliferation and differentiation. The generation of ENS neurospheres and networks in vitro made the ENSC culture suited for the investigation of the nutritional or neurotrophic requirements for ENSC development, especially neural network assembly, but it would not be possible without a chemically-defined culture system. Up to the 1950s, cells or tissues grown in media would simply die without the supplement of biological fluids, such as blood sera, amniotic fluids or embryo extracts [27], which contained a variety of nutrients, hormones, growth factors or even metabolic wastes. The use of biological fluids came along with the drawbacks of enigmatic chemical compositions, lot-to-lot variations, and cytotoxic substances [28], resulting in chemically-undefined and inconsistent media to preclude researchers from formulating the requisite nutrients, hormones or growth factors for cell growth and development. It was not until 1973 that primary DRG neurons were successfully grown under serum-free conditions [29]. The supplement of special growth-promoting constituents of hormones, trace elements or defined molecules (such as transferrin) [30,31,32] led to groundbreaking advancement in the cultivation of neural cells [27], making it feasible to bring primary CNS cells into serum-free cultures [31].

Over the past two decades, ENSC culture had proved achievable in serum-free NSM [11,12,33], but required SRM to be supplemented with CEE [2,17,19]. However, we demonstrated that SRM sufficed to support ENS network assembly in vitro, independently of CEE. Therefore, it was superfluous to supplement SRM with CEE in the context of neuronal differentiation and wiring. By selective elimination of one specific ingredient from CEE-free SRM, we were able to assess the influences of the removed component on neurogenesis, gangliogenesis, and neuronal wiring. ENS network assembly was demonstrated to have relevance to bFGF and B27 rather than IGF-1, NBM, RA, and N2 in CEE-free SRM. The blockade of gangliogenesis following bFGF or B27 elimination from CEE-free SRM, to a certain extent, mirrored the failure of enteric neurospherogenesis in NSM deprived of bFGF or B27, suggesting the biologically relevant effects of bFGF and B27 on enteric neural cell aggregates in the form of ganglia or neurospheres.

NBM and B27 were developed by Brewer et al. for growing hippocampal neurons [23] and had widespread neuronal cell applications. In this study, NBM showed no substantial benefit for ENS network assembly despite its capacity to promote the outgrowth of murine NCSCs [3]. N2 supplement included insulin, transferrin, progesterone, putrescine, and selenium [32], and was actually part of 20 components in B27 supplement [23]. Therefore, selective N2 elimination from CEE-free SRM merely reduced the dosages of N2 components in media, and had no influence on ENS network assembly. B27 was demonstrated to affect ENS network assembly in vitro. Among its 20 ingredients, we identified vitamin A derivatives of retinyl acetate (or RA) as crucial to ENS network assembly. RA is the well-known signaling molecule for neuronal patterning and differentiation in CNS development [34], and also essential for ENS development [35] with relevance to RA effects on accelerating ENSC specification from neural crest derivatives [36], and orchestrating their migration [37] and colonization to the gut [38]. Our results further disclosed the vital role of RA in ENS neuronal differentiation in the context of neurite outgrowth and network assembly. In the absence of vitamin A derivatives, enteric TUBB3^+^ neurons were mostly neuriteless and stagnated in a p75^high^ state. Since the p75 expression of developing enteric neurons declined as neuronal differentiation proceeded [25], these TUBB3^+^p75^high^ enteric neurons might represent the less mature phenotype, unfavorable for gangliogenesis and neuronal wiring. Notably, RA addition restored neuronal maturation toward TUBB3^+^p75^low^/TUBB3^+^p75^−^ neurons in the course of ENSC differentiation along a neuronal lineage, thereby promoting gangliogenesis, neurite outgrowth, and ENS network assembly. It was essentially in line with the promoting effects of RA on neuronal differentiation and neurite outgrowth of CNS [39,40] or DRG [34,41] stem cells, but in sharp contrast with the outcome reported by Sato et al., where RA promoted enteric neuronal differentiation but reduced neurite outgrowth [42]. Additional in-depth investigations into the influences of other B27 components on ENS network assembly may further define the signals that specify neuronal wiring, contributing to better understanding of the ENSC development.

Neural circuit formation relies upon the guidance of axonal growth cones that probe their surroundings for extracellular signals to mediate neurite outgrowth and synaptic connections. These signals involve different families of attractive or repulsive molecules, well-known as axon guidance cues [43]. There is compelling evidence that classic axon guidance cues regulate neural network assembly in collaboration with traditional growth factors [44]. Featuring the capacity of promoting stem cell proliferation, differentiation or survival, growth factors of bFGF, IGF-1, and EGF have been commonly supplemented into the protocols to propagate neural stem cells in neurosphere cultures. It was not until recent years that their later-phase role in neuronal morphogenesis, connectivity, and network assembly has increasingly attracted attention [44,45]. Both bFGF [46,47] and IGF-1 [48,49] were reported to promote CNS neurogenesis and neurite outgrowth. However, this study identified bFGF rather than IGF-1 as the potent neurotrophic factor for ENS network assembly in the context of promoting neurogenesis, gangliogenesis, and neuronal wiring despite the fact that IGF-1 supplementation might promote ENSC survivals [16]. These results were in keeping with the stimulatory effects of bFGF on neuronal differentiation and neurite outgrowth in neural crest-derived chromaffin cells [50], and also reflected the ENS morphological abnormalities of less neurons and smaller ganglia in bFGF-knockout mice [51]. As with bFGF, EGF was a potent mitogen [5] for CNS precursors [52] and exhibited neurotrophic activities of enhancing CNS neuronal survivals and neurite outgrowth in primary culture [53]. Although EGF might promote NCSC differentiation toward neurons and melanocytes [54], the loss of EGF receptors (EGFRs) contributed to massive axon outgrowth and branching of DRG neurons in *Egfr* null mutant mice [55]. This study further proved that EGF suppressed enteric neurogenesis, gangliogenesis, and network assembly. Together, they highlighted the inhibitory role of EGFR pathway in neurite outgrowth and branching in the later-phase development of NCSC-derived DRG and ENS neurons.

It has been well-recognized that EGF and bFGF affect neural network assembly directly by the alterations of growth cone cytoskeleton to modulate neurite extension [45,56], linked to invadopodia formation [57] or indirectly through non-neuronal cells, such as astrocytes [58], known as glia-growth cone interactions [59]. Moreover, neurite outgrowth might involve the interactions between cell adhesion molecules and EGF/FGF receptors [60,61]. Although it is clear now that both EGF [62] and bFGF [63] signal through receptor tyrosine kinases to trigger a variety of intracellular signaling cascades that regulate a series of biological processes in CNS or ENS development, the detailed molecular mechanisms underlying EGF or bFGF-mediated regulation of neurite guidance and outgrowth remain mostly shrouded in mystery [45] and await further experimental elucidation in the future. Notably, EGF exerted inhibitory effects on ENSC differentiation and development, not only in sharp contrast to its neurotrophic stimulation of cultured CNS neurons [53], but also diametrically opposed to bFGF, which was neurotrophic for ENSCs. It might explain the reason why ENSC-containing neurosphere isolation in SRM demanded EGF supplementation once neurosphere-like bodies appeared [17]. Therefore, the capacity of EGF to arrest enteric neuronal wiring made NSM particularly well-suited for ENSC enrichment in the form of neurospheres from enterocytes. Notably, isolated neurospheres in our laboratory could be maintained in NSM for at least 2–3 weeks without the loss of their capacity to assemble neural networks in SRM (unpublished data).

Energy metabolism plays an important role in neural stem cell function and fate [64]. The basal medium of SRM is low-glucose DMEM with 1.0 mg glucose/mL, whereas NSM contains glucose of 3.151 mg/mL (DMEM/F12 (1:1), (4.5 + 1.802)/2). High-glucose DMEM (4.5 mg/mL) had been used to prepare the formula for ENSC cultures [65]. Although nutrient media with higher glucose levels might hamper oxygen diffusion and enable anaerobic glycolysis with excessive lactate formation in cells, leading to neuronal cell damage [28], high-glucose SRM retained the capacity to assemble enteric neural networks in the bFGF- and B27-dependent patterns as observed in low-glucose SRM. Notably, high-glucose CEE-free SRM, comparing favorably in the gangliogenesis and neurogenesis with its low-glucose counterpart, remained ideally suited for ENS network assembly in vitro.

## 5. Conclusions

This study demonstrated the feasibility of assembling enteric neural networks in culture systems independently of complex and undefined biological fluids. The chemically-defined and consistent CEE-free SRM permitted us to investigate the nutritional or neurotrophic requirements for ENS network assembly. We identified bFGF and vitamin A as indispensable to the assembly of enteric ganglia and neural networks, whereas EGF, a CNS neurotrophic factor, proved inhibitory for enteric neurogenesis, gangliogenesis, and network assembly. Our results might help in better manipulating ENSC propagation and differentiation in vitro, and shedding light on the molecular pathways that regulate ENS network assembly.

## Figures and Tables

**Figure 1 cells-11-02841-f001:**
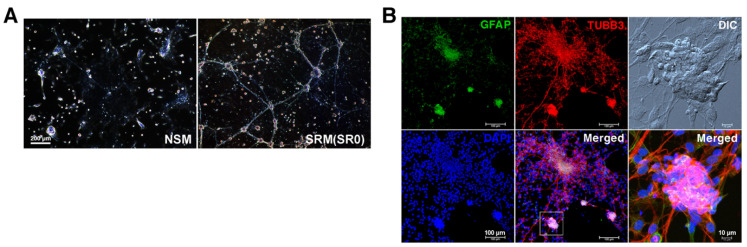
The formation of enteric neurospheres in NSM and SRM. Dispersed enterocytes (2 × 10^5^) from murine fetal gut were non-adherently nurtured in 6-well plates flooded with NSR or SRM. (**A**) On day 7, enteric neurospherogenesis was photographed live under a phase-contrast microscope. SRM supported the assembly of neural networks, whereas NSM generated discrete neurospheres with the scarcity of neurite outgrowth. (**B**) Neural networks assembled from dispersed enterocytes on coverslips in SRM-filled wells were subjected to immunofluorescence staining. The neurospheres, composed of TUBB3^+^ neurons and GFAP^+^ gliocytes, were structurally and morphologically linked to enteric ganglia and wired together by neuronal neurites to form neural networks (ganglionated plexuses). DIC: Differential interference contrast. SR0: Original SRM that contained CEE.

**Figure 2 cells-11-02841-f002:**
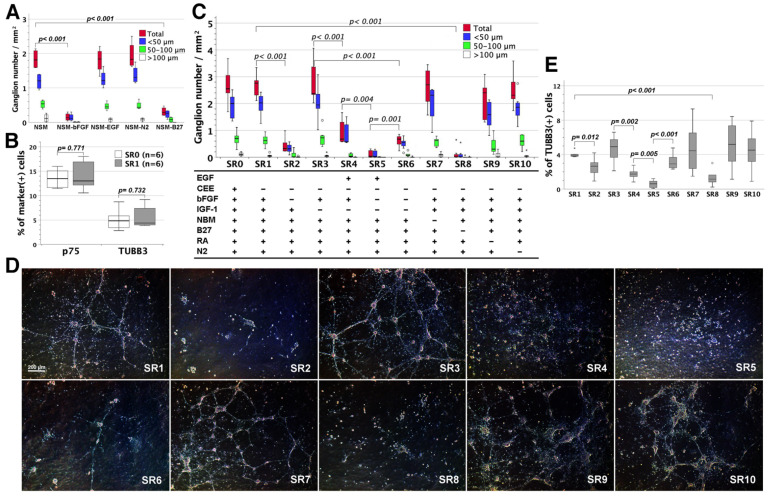
The influences of selective component elimination from NSM or SRM on the assembly of neurospheres or neural networks. Enterocytes (2 × 10^5^) were non-adherently grown in 6-well plates flooded with NSM, SRM or their modifications by selective component elimination. Enteric neurospheres (or ganglia) were enumerated and analyzed after 7-day cultivation. (**A**) Comparison was made to total neurosphere (neuriteless ganglion) number (the added numbers of neurospheres sized <50 μm, 50–100 μm, and >100 μm) between NSM and its indicated modifications (*n* = 4 for each group). Selective elimination of bFGF (NSM-bFGF) or B27 (NSM-B27) from NSM significantly reduced enteric neurospheres (*t*-test), whereas EGF or N2 omission (NSM-EGF or NSM-N2) made no change to neurospherogenesis. (**B**) Cultivated enterocytes in original SRM (SR0) and CEE-free SRM (SR1) were trypsinized, intracellularly stained with fluorescence-conjugated anti-p75 and anti-TUBB3, and analyzed by flow cytometry. SR0 and SR1 did not differ in the fractions of p75^+^ ENSCS and TUBB3^+^ neurons. (**C**) SRM was modified by selective elimination of one specific component or additional EGF supplementation as shown (*n* = 7 for each group). Comparison was made to total ganglion number of SR0 vs. SR1, SR1 vs. its indicated modifications (SR2, 3, 7–10) or two modified conditions that varied in only one supplement (SR3 vs. SR4, SR4 vs. SR5, SR5 vs. SR6, and SR2/SR3 vs. SR6). The *p*-values of <0.05 (*t*-test) were shown in the figure. Outliers (circle) and extremes (asterisk) were values of 1.5–3 times and more than 3 times interquartile ranges from the end of a box, respectively. (**D**) The images of ENSC development were taken live on day 7 in a representative set of experiments using the same batch of enterocytes. Day 7 cells in SRM and its relevant modifications (*n* = 6) were analyzed for TUBB3^+^ neurons by flow cytometry after intracellular staining. Comparison was made to TUBB3^+^ neuron fractions between the two groups as indicated above in (**C**).

**Figure 3 cells-11-02841-f003:**
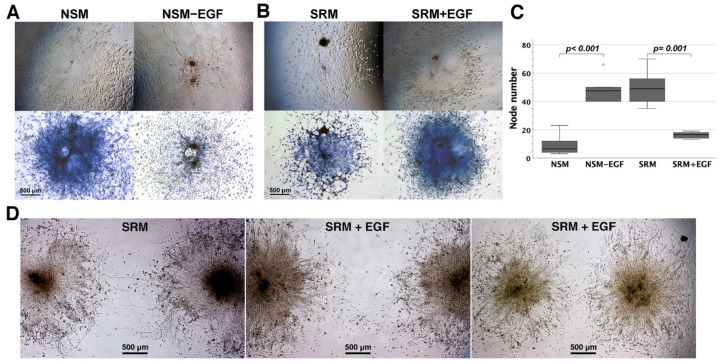
The EGF effects on network assembly of enteric neurospheres. Enteric neurospheres generated in NSM were nurtured adherently in NSM, EGF-free NSM (NSM-EGF), CEE-free SRM (SRM), and EGF-supplemented CEE-free SRM (SRM + EGF) for 1 week. (**A**) NSM-EGF gave rise to neural network assembly in preference to NSM. The networks were photographed live (upper) and after Diff-Quik staining (lower). (**B**) SRM + EGF downregulated neuronal wiring in contrast with extensive neuronal wiring in SRM. (**C**) The neuroconnectivity in the network assembled from a single neurosphere was graded by the amount of neurite intersections after Diff-Quik staining (*n* = 6 for each group). Groups NSM-EGF and SRM compared favorably in the intersection number with groups NSM and SRM + EGF, respectively. Outliers (circle) were values of 1.5–3 times interquartile ranges from the end of a box. (**D**) The neurite outgrowth and connectivity between two individual neurospheres in groups SRM and SRM + EGF were examined on day 7. SRM might help in wiring neurospheres together (left panel). However, SRM + EGF blocked neurite extension from neurospheres and led to the absence of connectivity between two independent neurospheres (middle panel) even though there was a closer distance between two neurospheres (right panel). The images shown were representative of three independent experiments.

**Figure 4 cells-11-02841-f004:**
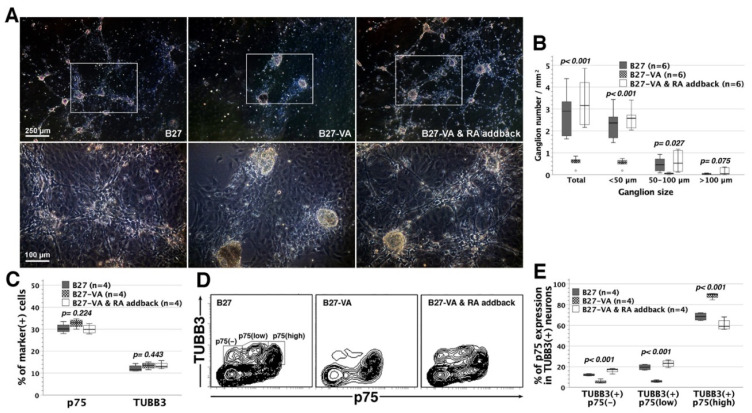
The effects of vitamin A derivatives on neuronal differentiation and wiring. As a modified version of the original B27 without vitamin A (retinyl acetate), B27 minus vitamin A (group B27-VA) was substituted for B27 to prepare RA/CEE-free SRM (SR9) that was completely free from vitamin A derivatives. (**A**,**B**) Group B27-VA had reduced gangliogenesis and lost the ability to support neuronal wiring, as compared with SR9 supplemented with original B27 (group B27), whereas RA addback to group B27-VA (group B27-VA & RA addback) restored gangliogenesis (ANOVA) and neuronal wiring. Outliers (circle) were values of 1.5–3 times interquartile ranges from the end of a box. (**C**) Flow cytometric analyses showed no significant difference in the fractions of p75^+^ ENSCs and TUBB3^+^ neurons among the three groups. (**D**) Contour plots addressed the variations of p75 expression in TUBB3^+^ enteric neurons of the three groups. (**E**) Group B27-VA exhibited significantly less TUBB3^+^p75^−^ and TUBB3^+^p75^low^, but more TUBB3^+^p75^high^ neurons than group B27. Group B27-VA & RA addback recovered the levels of TUBB3^+^p75^−^, TUBB3^+^p75^low^, and TUBB3^+^p75^high^ neurons.

**Figure 5 cells-11-02841-f005:**
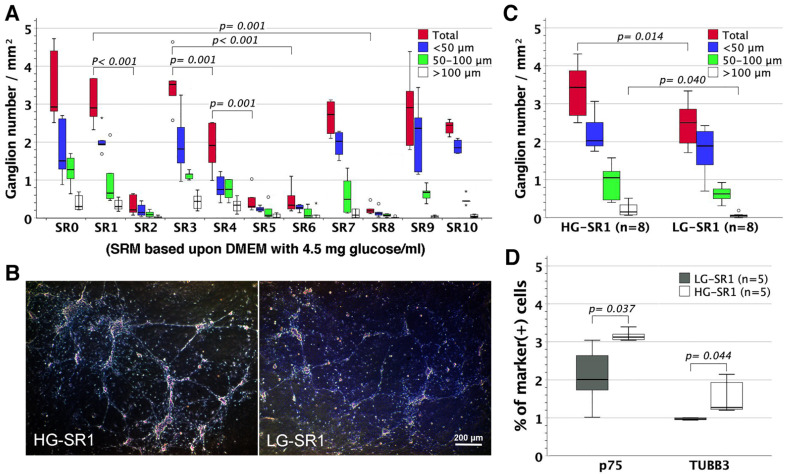
Neural network assembly in high-glucose SRM and its relevant modifications. High-glucose (4.5 mg/mL) DMEM was substituted for low-glucose (1.0 mg/mL) DMEM to make up high-glucose SRM. Outliers (circle) and extremes (asterisk) were values of 1.5–3 times and more than 3 times interquartile ranges from the end of a box, respectively. (**A)** High-glucose SRM (*n* = 6 for each group) was modified in accordance with the rule previously used for low-glucose SRM in Figure 2C. The selective elimination of bFGF or B27 from high-glucose SRM proved to be most inimical to neural network assembly, identical to what was observed in low-glucose SRM. (**B**) High-glucose CEE-free SRM (HG-SR1) supported neural network assembly similar to low-glucose CEE-free SRM (LG-SR1). (**C**) HG-SR1 generated more ganglia than LG-SR1, especially ganglion size of >100 μm. (**D**) HG-SR1 compared favorably in the fraction of p75^+^ ENSCs and TUBB3^+^ neurons with LG-SR1.

## Data Availability

All data are contained in this article.

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
