# Peer review of "Enteric Neural Network Assembly Was Promoted by Basic Fibroblast Growth Factor and Vitamin A but Inhibited by Epidermal Growth Factor"

_cells, 2022, doi:10.3390/cells11182841_

Round 1
Reviewer 1 Report
The findings of this study are interesting and can be of significant interest to the wide readership of Cells. The authors may be encouraged to improve the description of the results. Portions of the discussion read as being speculative and not backed by experimental data. These concerns need to be addressed.
Author Response
The findings of this study are interesting and can be of significant interest to the wide readership of Cells. The authors may be encouraged to improve the description of the results. Portions of the discussion read as being speculative and not backed by experimental data. These concerns need to be addressed.
R: We have rewritten our experimental findings and revised the discussion section, based upon the data obtained in this study. If the discussion pertaining to the effects of EGF, bFGF and RA on the neurogenesis, gangliogenesis and neural network assembly was not backed by our experimental findings, we refer it to the articles already published in the literature, and list them in the reference section.
Reviewer 2 Report
Jeng-Chang Chen and co-authors have presented the manuscript "Enteric Neural Network Assembly Was Promoted by Basic Fibroblast Growth Factor and Vitamin A but Inhibited by Epidermal Growth Factor". This has been a pleasurable review with very few minor corrections to be made. In its current form, this paper is acceptable for publication.
If I had to be really pedantic and find something bothersome to fix would be to increase the contrast in microscopy images 1A, 2D, 4A and 5B. Increasing the contrast may help with clarity.
Author Response
Jeng-Chang Chen and co-authors have presented the manuscript "Enteric Neural Network Assembly Was Promoted by Basic Fibroblast Growth Factor and Vitamin A but Inhibited by Epidermal Growth Factor". This has been a pleasurable review with very few minor corrections to be made. In its current form, this paper is acceptable for publication.
If I had to be really pedantic and find something bothersome to fix would be to increase the contrast in microscopy images 1A, 2D, 4A and 5B. Increasing the contrast may help with clarity.
R: The contrast of all the images taken by a phase-contrast microscopy was adjusted to enhance the clarity as requested.
Reviewer 3 Report
The manuscript by Chen et al investigates the effect of different media and additives on neurogenesis, gangliogenesis and neuronal wiring in cultured enterocytes from murine fetal gut.
They found that FGF2 and vitamin A promoted, whereas EGF inhibited neuronal wiring.
The study is well written and the experiments and their outcomes are clearly described but overall a bit limited and superficial.
The results are relevant for researchers using cultured enteric neuronal stem cells but questions of broader scientific interest such as the molecular basis of the opposing effects of FGF2 and EGF are left unexplored.
FGF2 and EGF can entertain an overlapping set of signaling pathways. To enhance the interest and significance of the study, the authors should investigate whether engagement of different signaling pathways downstream of the receptors, different kinetics of activation or other molecular differences explain the opposing roles of these two growth factors.
Minor: Figures 2, 4 and 5 are partially too small and hard to read in a printout.
Author Response
The manuscript by Chen et al investigates the effect of different media and additives on neurogenesis, gangliogenesis and neuronal wiring in cultured enterocytes from murine fetal gut. They found that FGF2 and vitamin A promoted, whereas EGF inhibited neuronal wiring. The study is well written and the experiments and their outcomes are clearly described but overall a bit limited and superficial. The results are relevant for researchers using cultured enteric neuronal stem cells but questions of broader scientific interest such as the molecular basis of the opposing effects of FGF2 and EGF are left unexplored. FGF2 and EGF can entertain an overlapping set of signaling pathways. To enhance the interest and significance of the study, the authors should investigate whether engagement of different signaling pathways downstream of the receptors, different kinetics of activation or other molecular differences explain the opposing roles of these two growth factors.
R: The development of enteric neural stem cell (ENSC) culture systems mainly took advantage of the knowledge and protocols already available in CNS or dorsal root ganglion (DRG) stem cell biology. However, no general consensus has so far been reached on an optimal protocol for the isolation or propagation of enteric neurospheres. ENSC culture systems varied significantly from laboratory to laboratory (Development 2003, 130, 6387-6400; J Clin Invest 2013, 123, 1182-1191; Pediatr Surg Int 2003, 19, 340-344 & Neurogastroenterol Motil 2016, 28, 498-512) and even from report to report in the same research group (Gastroenterology 2009, 137, 2063-2073 e2064 & J Neurochem 2007, 103, 2665-2678). It led to not only a certain level of confusion over how to select a protocol in getting started on ENSC studies, but also the variations in resulting cell compositions to preclude unbiased comparisons across studies and a better evaluation for the effects of individual protocol components on enteric neurogenesis. Neurosphere medium (NSM) and self-renewal medium (SRM) represent the two widely-used but component-distinct formulas in ENSC applications. Using NSM and SRM, we investigated the nutritional or neurotrophic requirements for the expansion and development of ENSCs. NSM proved suitable for mass production of enteric neurospheres, whereas SRM turned out to be the excellent ENSC differentiation medium, capable of driving neuronal differentiation and wiring. Strikingly, we identified bFGF and vitamin A as indispensable to the assembly of enteric ganglia and neural networks, but EGF as inhibitory for enteric neurogenesis, gangliogenesis and network assembly. The functional distinction between NSM and SRM in ENSC cultures could be ascribed to the inhibitory effects of EGF on ENSC development.
As the reviewers were aware, these novel findings were of interest in the field of ENS stem cell biology, and would be even more if the molecular events underlying the opposite effects of EGF and bFGF on neurite outgrowth of enteric neurons could be further elucidated. However, such an investigation was far beyond the scope of this study so the authors did not go further into the molecular events as to how EFG and bFGF signal through receptor tyrosine kinases to regulate cell signaling and the cytoskeleton, and influence axon extension and network assembly. We are reporting the findings of this study in its current form at the prospect of their rapid sharing or communication in this field so as to expedite the progress of in vitro ENSC propagation.
To address the reviewer’s concern, we bring into discussion the up-to-date knowledge of molecular biology relevant to the regulation of axon guidance and outgrowth by EGF and bFGF (Lines 370-379 & 389-406). As expected, the detailed molecular mechanisms underlying FGF or bFGF-mediated regulation of neurite guidance and outgrowth remain mostly shrouded in mystery. It’s also worth mentioning that EGF receptor pathway, well-recognized as CNS-neurotrophic, has the inhibitory role in PNS neurite outgrowth and branching in the later-phase development of neural crest-derived DRG (J Invest Dermatol 2009, 129, 690-698) and ENS neurons (as shown in this study).
Minor: Figures 2, 4 and 5 are partially too small and hard to read in a printout.
R: Figures 2, 4 and 5 are amended to enlarge the texts as requested.
Reviewer 4 Report
This is a very nicely done study, assessing a topic of importance to neural crest and enteric nervous system researchers. While the work will likely not be of interest to a broader audience, the study certainly is deserving of publication and will be valuable to scientists who work with ex vivo models of enteric neurogenesis. The text should be edited slightly for clarity in English, for example the sentence on lines 159-60 that reads "DMEN/F12-based NSM supplemented with bFGF, EGF, N2 and B27 was serum-free, chemically-defined nutrient medium" is somewhat unclear. Figures 2-5 also have very small text that is difficult to read.
The authors focus largely on the effects of various media and additives on neurite outgrowth and network assembly. However, many scientists may actually desire media that inhibit terminal neuronal differentiation and that preserve cells a stem or progenitor state. Studies it assess this are probably outside the scope of this paper, and lack of such analysis is not a reason to oppose publication. However, it would be useful for the authors to discuss this in their introduction and discussion sections.
Overall, this is a well done study with clear significance for the neural crest and enteric nervous system fields.
Author Response
This is a very nicely done study, assessing a topic of importance to neural crest and enteric nervous system researchers. While the work will likely not be of interest to a broader audience, the study certainly is deserving of publication and will be valuable to scientists who work with ex vivo models of enteric neurogenesis. The text should be edited slightly for clarity in English, for example the sentence on lines 159-60 that reads "DMEN/F12-based NSM supplemented with bFGF, EGF, N2 and B27 was serum-free, chemically-defined nutrient medium" is somewhat unclear. Figures 2-5 also have very small text that is difficult to read.
R: We have revised the result and discussion sections to improve the clarity. The indicated sentence is now revised to read: “DMEN/F12-based NSM was supplemented with bFGF, EGF, N2 and B27 as a kind of serum-free and chemically-defined nutrient medium.” Figures 2-5 are also amended to enlarge the texts as requested.
The authors focus largely on the effects of various media and additives on neurite outgrowth and network assembly. However, many scientists may actually desire media that inhibit terminal neuronal differentiation and that preserve cells a stem or progenitor state. Studies it assess this are probably outside the scope of this paper, and lack of such analysis is not a reason to oppose publication. However, it would be useful for the authors to discuss this in their introduction and discussion sections.
Overall, this is a well done study with clear significance for the neural crest and enteric nervous system fields.
R: EGF was not an essential ingredient for self-renewal medium (SRM) but rather a requisite for neurosphere medium (NSM). This study characterized NSM as ideal for neurosphere enrichment from dispersed enterocytes, but SRM as excellent in driving neuronal differentiation and wiring. This distinction between NSM and SRM could be ascribed to EGF, which proved inhibitory for neurite extension and network assembly of ENS. In 2003, enteric neurospheres were first generated from dispersed enterocytes in SRM. At that time, EGF was supplemented into SRM once neurosphere-like bodies showed up without any reason given (Development 2003, 130, 6387-6400). It is now clear that EGF might exert its inhibitory effects on enteric neurogenesis, gangliogenesis and neuritogenesis. Of note, we found that NSM did have the capacity to preserve ENSCs in a progenitor state because isolated neurospheres in our laboratory could be maintained in NSM for at least 2-3 weeks without the loss of their capacity to assemble neural networks in SRM. We address the reviewer’s concern by describing these notions in the discussion section (Lines 409-415).
Round 2
Reviewer 3 Report
I still think that dissecting - at least to some degree - the molecular events downstream of EGFR versus FGFRs would have significantly increased the overall merit of the curent manuscript, but can accept the authors explanation that this would be beyond the scope of the their investigation. At least the authors have added some discussion of the matter. It remains puzzling that such an important issue has not received more attention in the neural stem cells community.